# Protective Effects of Ethanol Extract of Brazilian Green Propolis and Apigenin against Weak Ultraviolet Ray-B-Induced Barrier Dysfunction via Suppressing Nitric Oxide Production and Mislocalization of Claudin-1 in HaCaT Cells

**DOI:** 10.3390/ijms221910326

**Published:** 2021-09-25

**Authors:** Yuta Yoshino, Kana Marunaka, Mao Kobayashi, Haruka Matsunaga, Shokoku Shu, Toshiyuki Matsunaga, Akira Ikari

**Affiliations:** 1Laboratory of Biochemistry, Department of Biopharmaceutical Sciences, Gifu Pharmaceutical University, Gifu 501-1196, Japan; yoshino-yu@gifu-pu.ac.jp (Y.Y.); 136033@gifu-pu.ac.jp (K.M.); 155032@gifu-pu.ac.jp (M.K.); 165077@gifu-pu.ac.jp (H.M.); 165041@gifu-pu.ac.jp (S.S.); 2Education Center of Green Pharmaceutical Sciences, Gifu Pharmaceutical University, Gifu 502-8585, Japan; matsunagat@gifu-pu.ac.jp

**Keywords:** claudin, keratinocyte, UVB, nitric oxide, apigenin, flavonoid

## Abstract

Once weak ultraviolet ray-B (UVB) irradiates the skin cells, the generation of reactive nitrogen species (RNS), but not reactive oxygen species (ROS), is stimulated for the mislocalization of claudin-1 (CLDN1), an essential protein for forming tight junctions (TJs). Since our skin is constantly exposed to sunlight throughout our lives, an effective protection strategy is needed to maintain the skin barrier against weak UVB. In the present study, we investigated whether an ethanol extract of Brazilian green propolis (EBGP) and flavonoids had a protective effect against weak UVB irradiation-induced barrier dysfunction in human keratinocyte-derived HaCaT cells. A pretreatment with EBGP suppressed TJ permeability, RNS production, and the nitration level of CLDN1 in the weak UVB-exposed cells. Among the propolis components, apigenin and apigenin-like flavonoids have potent protective effects against NO production and the mislocalization of CLDN1 induced by UVB. The analyses between structures and biological function revealed that the chemically and structurally characteristic flavonoids with a hydroxyl group at the 4′ position on the B-ring might contribute to its protective effect on barrier dysfunction caused by weak UVB irradiation. In conclusion, EBGP and its component apigenin protect HaCaT cells from weak UVB irradiation-induced TJ barrier dysfunction mediated by suppressing NO production.

## 1. Introduction

As part of the skin’s barrier function, tight junctions (TJs) play a critical role in maintaining the homeostasis of electrolyte ions and water in the tissue surface. The TJ barrier strand is composed of transmembrane proteins, including claudin (CLDNs) and occludin, scaffold proteins, zonula-occludens, and cytoskeletal actin, which tightly close the cell−cell contact area [1]. Since these reports indicate that the expression level of CLDN1 decreased in the skin with atopic dermatitis [2] and CLDN1 knockout mice showed skin barrier dysfunction [3], CLDN1 is considered as a main component for the TJ barriers in the skin among over 20 subtypes of CLDNs. Our recent studies suggest the expression level and the cellular localization of CLDN1 might be related closely to the integrity of the TJ barrier in the skin [4,5].

A severe level of ultraviolet ray-B (UVB) irradiation causes reactive oxygen species (ROS), reactive nitrogen species (RNS), and DNA damage, resulting in the dysfunction of the skin barrier and an elevation of risk factors for skin cancer [6]. In contrast, non-cytotoxic “weak” UVB irradiation stimulates RNS generation, but not ROS, leading to barrier dysfunction associated with the mislocalization of CLDN1 in our previous report [5]. Since our skin is constantly exposed to sunlight throughout our lives, an effective protection strategy is needed to maintain skin health against weak UVB. However, there are no effective dietary supplements to protect the skin against RNS-related barrier dysfunction. Therefore, effective compounds or natural ingredients that constantly protect the skin from UVB-induced stresses may become useful ingredients for skin care products.

Propolis is a resinous bee product, composed of floral bud and jelly, which has diverse and potent biological activities [7]. In recent decades, much research has shown that the natural products derived from propolis have potential benefits for the skin, such as protective effect against UV-induced skin damage or sunburn by potent antioxidative and anti-inflammatory effects [8,9]. Therefore, propolis products are attracting much attention as useful candidates for skin health care. Among the propolis products, an ethanol extract of Brazilian green propolis (EBGP), which contains various kinds of bioactive compounds, such as flavonoids, phenolic acids, esters, terpenoids, and cinnamic acid derivatives, is well known to have multiple physiological functions including antibacterial, anti-inflammation, antioxidant, and antitumor effects [10,11,12,13]. Moreover, we recently reported that EBGP had a protective effect against oxidative stress-induced skin barrier dysfunction on human keratinocyte-derived cells via its potent antioxidative effect [4], suggesting the utility of the ingredient for skin health. However, it remains unclear whether EBGP has a protective effect for the skin barrier without antioxidative activity.

In the present study, we investigated whether EBGP and its components had a protective effect against weak UVB irradiation-induced barrier dysfunction in human keratinocyte-derived HaCaT cells. Moreover, the relationship between the structural character and the biological functions of each flavonoid and phenylpropanoid were also demonstrated. The present study can provide useful information about the structure and function of natural product derivatives, leading to the development of dietary supplements or cosmetic ingredients for skin care in the future.

## 2. Results

### 2.1. Protective Effect of EBGP on Weak UVB-Induced Barrier Dysfunction

We recently reported that weak UVB irradiation transiently induces mislocalization of CLDN1, resulting in the defect of TJ barrier function [4]. The protective effect of EBGP on barrier function were investigated using the cells cultured on transwell plates. Pretreatment with EBGP significantly suppressed weak UVB-induced decrease in transepithelial electrical resistance (TER), an indicator of electrolyte ion permeability (Figure 1A). Furthermore, an increase of lucifer yellow (LY) flux, an indicator of small molecular permeation, was significantly suppressed by EBGP in a dose-dependent manner (Figure 1B). These results suggest that EBGP has protective effect against weak UVB-induced TJ barrier dysfunction.

### 2.2. Protective Effect of EBGP on Weak UVB-Induced RNS Generations

Once the weak UVB irradiated the cells, the generation of reactive nitrogen species (RNS) was stimulated, which induced mislocalization of CLND1 [5]. Next, the effects on RNS generation were investigated using specific probes for Ca^2+^ influx, nitrogen oxide (NO) production, and peroxynitrite production, which is a more reactive form of RNS. The pretreatment with EBGP significantly suppressed the weak UVB-induced increase in Ca^2+^ influx and production of both NO and peroxynitrite (Figure 2). These results suggest that EBGP has protective effects against weak UVB-induced RNS generations in the cells.

In the epidermal TJ barrier, the cellular localization and function of CLDNs are strictly regulated by the status of its post-translational modifications [14]. Especially, tyrosine nitration relates to the mislocalization of CLDN1 in HaCaT cells [5]. Thus, the level of tyrosine-nitrated CLDN1 protein was evaluated by immunoprecipitation assay using anti-nitro-tyrosine and anti-CLDN1 antibodies. The pretreatment with EBGP significantly suppressed the UVB-induced elevation of tyrosine-nitrated CLDN1 protein (Figure 3A). Intracellular NO is produced from L-arginine by the NO synthase (NOS) family [15]. The expression level of NOS3 protein was increased by UVB irradiation and suppressed by pretreatment with EBGP (Figure 3B). These results are consistent with its suppression effects on RNS generation in Figure 2. These results indicate that EBGP suppresses NO production and RNS-related post-translational modification of CLDN1 protein caused by weak UVB irradiation.

### 2.3. The Effects of EBGP Components on RNS Generation

Polyphenols, cinnamic acid derivatives and flavonoids are well known as biologically active components of propolis. Since EBGP showed potent protective effects against UVB irradiation-induced RNS generation, we examined the effects of propolis components on UVB-induced RNS generation. Among the compounds examined, the treatment with apigenin completely suppressed weak UVB-induced Ca^2+^ flux and production of both NO and peroxynitrite (Figure 4A). However, neither caffeic acid phenethyl ester (CAPE), which is a polyphenol, nor artepillin C, which is a cinnamic acid derivative, suppressed UVB-induced RNS generation. These results suggest that flavonoids might be the main compound suppressing RNS generation among propolis components. A lower dose (1 μM) of propolis components did not suppress the UVB-induced Ca^2+^ flux. (Figure 4B). The contents of the compounds in EBGP (10 μg/mL) are less than 1 μM of compound [16]. Therefore, the protective effect of EBGP might be derived from the combined effects, but not the effect of a single compound.

### 2.4. Chemical Structures and Biological Activity in Flavonoids

Since apigenin completely protected cellular barrier function against weak UVB-induced RNS generation, other flavonoids might also have similar protective effects. Among the flavonoids, we selected those with a structure similarly to apigenin and examined their effects on NO production and CLDN1 mislocalization. Quercetin suppressed both UVB-induced NO production (Figure 5A) and CLDN1 mislocalization (Figure 5B). On the other hand, kaempferol and kaempferide tended to suppress NO production, but not significantly. Kaempferol suppressed the mislocalization of CLDN1. Chrysin and apiin, which is a glycoside of apigenin, did not have these effects. These results indicate that flavonoids might be the main active compounds and apigenin has the strongest effects among the flavonoids. Therefore, the next step is to investigate the effects of apigenin in more detail.

### 2.5. Protective Effect of Apigenin on UVB-Induced Barrier Dysfunction

In our previous study, weak UVB irradiation induced the mislocalization of CLDN1, but did not decrease the expression level of the protein [5]. Treatment with apigenin protected against the UVB irradiation-induced mislocalization of CLDN1 (Figure 6A). Interestingly, *N*^G^-nitro-L-arginine methyl ester (L-NAME), an inhibitor of NO synthesis, also completely suppressed the mislocalization of CLDN1 caused by UVB irradiation (Figure 6A), indicating that intracellular NO production may be a critical response to maintain homeostasis of CLDN1 on the TJ area. In the transwell assay, pretreatment with apigenin, but not CAPE, suppressed a UVB-induced decrease in TER (Figure 6B) and an increase in the paracellular flux of LY (Figure 6C). Moreover, the level of tyrosine-nitrated CLDN1 protein was decreased by apigenin (Figure 7A). The expression level of NOS3 protein was decreased by pretreatment with apigenin prior to UVB irradiation (Figure 7B). These results suggest that apigenin has a potent protective effect against UVB-induced barrier dysfunction by suppressing RNS generation.

## 3. Discussion

In the present study, the EGBP and some flavonoids showed significant protective effects against weak UVB irradiation-induced barrier dysfunction by suppressing the NO-related RNS signal in HaCaT cells. EBGP is composed of various ingredients from various plants. Since the effects of apigenin were similarly to those of EBGP on UVB-induced barrier dysfunction, apigenin may be one of active components. We previously reported that UVB-irradiation is sensed by the opsin-2 protein, leading to transient receptor potential cation channel subfamily V member 1 (TRPV1) activation-related Ca^2+^ influx to cytoplasm [5]. The UVB-induced elevation of intracellular free Ca^2+^ concentration was suppressed by both EBGP and apigenin (Figure 2 and Figure 4), suggesting that the UVB-sensed TRPV1 Ca^2+^ channel is involved in these protection mechanisms (Figure 8). However, the detail of the molecular mechanism remains unclear. Further analyses are needed to get more useful chemical structural information of the flavonoids.

Flavonoids are phenolic compounds isolated from various plants and most of them have potent antioxidant activity. Many research groups have reported the antioxidant activity of flavonoids, based on their ability to reduce free radical formation and to scavenge free radicals [17]. In addition to antioxidant activity, some flavonoids are thought to interact with specific enzymes or ion channels. The flavonoids of *Polygonum orientale* L. had an anti-rheumatoid arthritis effect in a rat model and the interaction of flavonoids with NOS protein as a target protein was demonstrated in the molecular docking study [18]. Eriodictyol, a flavonoid, acts as an antagonist of the TRPV1 receptor to induce antinociception [19]. These reports indicate that flavonoids might interact with the NOS protein or TRPV1 Ca^2+^ channel, up to their specific chemical structural property.

The structural character of flavonoids, having more than just hydroxyl residue on their B-ring position, was reported to relate to their radical scavenging activity [20,21,22]. Similar to these reports, our study revealed that the suppressing effects on weak UVB-induced RNS generation were shown on the B-ring hydroxylated flavonoids (apigenin and quercetin), but not other flavonoids (chrysin, kaempferide and apiin). Kaempferol, also a B-ring hydroxylated flavonoid, similarly tended to suppress NO production. Chrysin does not have a hydroxyl group on the B-ring compared with apigenin. In the structures of kaempferide and apiin, the B-ring hydroxyl residue is methylated and O-glycosylated, respectively. These results indicate that the hydroxyl residue, especially at the 4′ position on the B-ring, in flavonoids may be an essential structure to interact with some target protein, which has a crucial role for RNS signaling in weak UVB-irradiated cells (Figure 9). Moreover, apiin, an O-glycosylated form of apigenin, did not show a protective effect against weak UVB, suggesting that the glycoside-formed flavonoids might not penetrate the cell membrane into the cytosol because of their chemical polarity. However, since the target proteins of flavonoids in cells are still unknown, further study is needed to discover what protein is the main target of these flavonoids. Structurally characterized B-ring-hydroxylated flavonoids might become useful candidates for skin care ingredients or dietary supplements aimed at protecting skin health from UVB.

Apigenin presented a potent effect on UVB-induced RNS generation (Figure 4). There are various reports on the effects of apigenin on RNS. In another study, apigenin and its derivatives also suppressed lipopolysaccharide-induced NO production in mouse macrophage RAW264.7 cells [23,24]. On the other hand, there are some reports of the opposite effect of apigenin on NO production. In the endothelial cells, apigenin promotes NOS activation and NO production [25]. Apigenin promotes antibacterial activity by NO production [26]. These reports indicate that the effect of apigenin on NO regulation might be different depending on the tissues or the cell types.

Andrade et al., [16] have reported on the contents of phenolic and flavonoid compounds in green propolis by ultra-high-performance liquid chromatographic system coupled with tandem mass spectrometry (artepillin C; 4.80 ± 0.03 mg/g, CAPE; 0.29 ± 0.02 mg/g, apigenin; 0.01 ± 0.00 mg/g). The concentration of EBGP (10 μg/mL) was equal to approximately 0.16 μM of artepillin C, 0.016 μM of CAPE, and 0.0037 μM of apigenin, respectively. Although the content of flavonoids such as apigenin in EBGP is low, EBGP contains a wide variety of flavonoids. Thus, the potent protective effect of EBGP might be due to the combined effect of the flavonoids with characteristic chemical structures.

The function of NO on skin health is controversial. The expression levels of NOS3 and NO production have been reported to be necessary for aryl hydrocarbon receptor-ligand-mediated keratinocyte differentiation [27]. In the macrophage, NO is required to possess antibacterial activity for the immune system [26]. In the present study, we demonstrated that an overproduced NO-related signal mediated the UVB-induced mal effect for the skin barrier as RNS. The more detailed functions of NO and NO-related signals in the skin need to be clarified in a future study.

In conclusion, EBGP and its component apigenin protected HaCaT cells from weak UVB irradiation-induced TJ barrier dysfunction, mediated by suppressing NO and RNS generation.

## 4. Materials and Methods

### 4.1. Materials

Rabbit anti-CLDN1 and Alexa Fluor 555 anti-rabbit antibodies were obtained from Thermo Fisher Scientific (San Diego, CA, USA). Goat anti-β-actin antibody was obtained from Santa Cruz Biotechnology (Santa Cruz, CA, USA). L-NAME, LY, NiSPY-3, Quest Fluo-8 AM, rabbit anti-nitrotyrosine, and rabbit anti-NOS3 antibodies were from Dojindo Laboratories (Kumamoto, Japan), Biotium (Fremont, CA, USA), Goryo Kagaku (Hokkaido, Japan), AAT Bioquest (Sunnyvale, CA, USA), R&D Systems (Minneapolis, MN, USA), and Cell Signaling Technology (Beverly, MA, USA), respectively. DAF-2DA, CAPE, apigenin, kaempferol, kaempferide, chrysin, quercetin, and apiin were from Cayman Chemical (Ann Arbor, MI, USA). EBGP was kindly provided from Yamada Bee Company (Okayama, Japan).

### 4.2. Cell Cultures

Human skin derived-immortal keratinocyte cell line, HaCaT cells [28] were maintained in Dulbecco’s modified Eagle’s medium (Sigma-Aldrich, Saint Louis, MO, USA) supplemented with 5% fetal bovine serum (FBS; Sigma-Aldrich, Saint Louis, MO, USA), 0.07 mg/mL penicillin-G potassium, and 0.14 mg/mL streptomycin sulfate in a 5% CO_2_ atmosphere at 37 °C. The cells were passed using 0.5% trypsin every 3–4 days. EBGP stock solution (10 mg/mL in 100% ethanol) was dissolved in DMEM to prepare the experimental medium (5, 10 μg/mL). As the vehicle, 0.1% ethanol containing DMEM was used. In the other experiments, we used 0.1% dimethyl sulfoxide as the vehicle of propolis components and the other reagents.

### 4.3. UVB Irradiation

UVB irradiation was carried out as previously described [5]. Using UV Crosslinker CL-1000M (Analytik Jena, Upland, CA, USA), UVB peaking wavelength at 302 nm was emitted on the cells for approximately 7 s at 5 mJ/cm^2^. After 6 h from UVB irradiation, the cells were used for the experiments. Since the cell viability was not affected by 5 mJ/cm^2^ UVB irradiation for 24 h in previous study [4], this non-cytotoxic dose was used as “weak” UVB.

### 4.4. Confocal Microscopy

The cells were cultured on cover glasses in 35 mm dishes (70,000 cells/dish) for 3 days. After 6 h of UVB irradiation, the cells were fixed with methanol at −30 °C for 15 min. Then, the cells underwent immunocytochemistry assay using rabbit anti-CLDN1 antibody (1:100 dilution) and Alexa Fluor 555 secondary antibody. The fluorescence images were taken near the apical membrane using an LSM700 confocal microscope (Carl Zeiss, Jena, Germany) as described previously [4].

### 4.5. Paracellular Permeability Assay

The cells (5,000 cells/well) were cultured on Transwell plates for 3 days (0.4 μm pore size, Corning, NY, USA). After 6 h of UVB irradiation, TER and the paracellular permeability to LY, a fluorescent paracellular flux marker, were measured as previously described [4]

### 4.6. Sodium Dodecyl Sulfate-Polyacrylamide Gel Electrophoresis and Immunoblotting

The cells (70,000 cells/dish) were cultured on 35 mm dishes for 3 days. After 6 h from UVB irradiation, the cells were collected. In immunoprecipitation assay, we incubated the cell lysates with anti-CLDN1 antibody and protein G-Sepharose beads for 16 h at 4°C with gentle rocking. Sodium dodecyl sulfate-polyacrylamide gel electrophoresis (SDS-PAGE) and immunoblotting were performed as previously described [4]. Band density was quantified using ImageJ software (National Institute of Health software). The signals were normalized using the β-actin as loading control. In immunoprecipitation assay, the data were normalized to the signals by CLDN1 as control.

### 4.7. Intracellular RNS Contents and Free Ca^2+^ Concentration

The cells (7000 cells/well) were cultured on 96-well plates for 2 days. The cells were incubated with fluorescent probes for 30 min; DAF-2DA, a fluorescent indicator of NO; NiSPY-3, a fluorescent peroxynitrite indicator; and Fluo-8 AM, a fluorescent Ca^2+^ indicator, respectively. After 3 h of UVB irradiation, the fluorescence intensities were measured using an Infinite F200 Pro microplate reader (Tecan, Switzerland).

### 4.8. Statistical Analysis

The data are presented as means ± standard error of the mean. Differences between groups were analyzed using one-way analysis of variance, and corrections for multiple comparison were made using Tukey’s multiple comparison test. Comparisons between two groups were made using Student’s *t*-test. Statistical analyses were performed using KaleidaGraph version 4.5.1 software (Synergy Software, Reading, PA, USA). Differences were assumed as significant at *p* < 0.05.

## Figures and Tables

**Figure 1 ijms-22-10326-f001:**
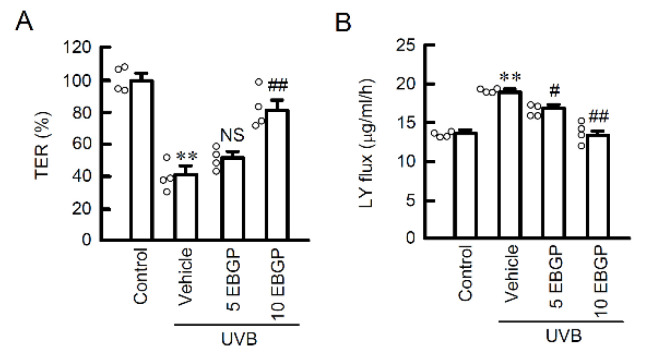
Effect of EBGP on TJ permeability against weak UVB irradiation. The cells were cultured on transwell plates and incubated in the absence (vehicle) and presence of 5 or 10 μg/mL EBGP for 30 min, followed by exposure to 5 mJ/cm^2^ UVB for 6 h. (**A**) TER was measured by volt-ohm meter. The data were presented as a percentage of control. (**B**) Paracellular permeability of LY was measured using a fluorescence spectrometry. *n* = 4. ** *p* < 0.01 vs. control. ^#^ *p* < 0.05, ^##^ *p* < 0.01 and NS *p* > 0.05 vs. vehicle.

**Figure 2 ijms-22-10326-f002:**
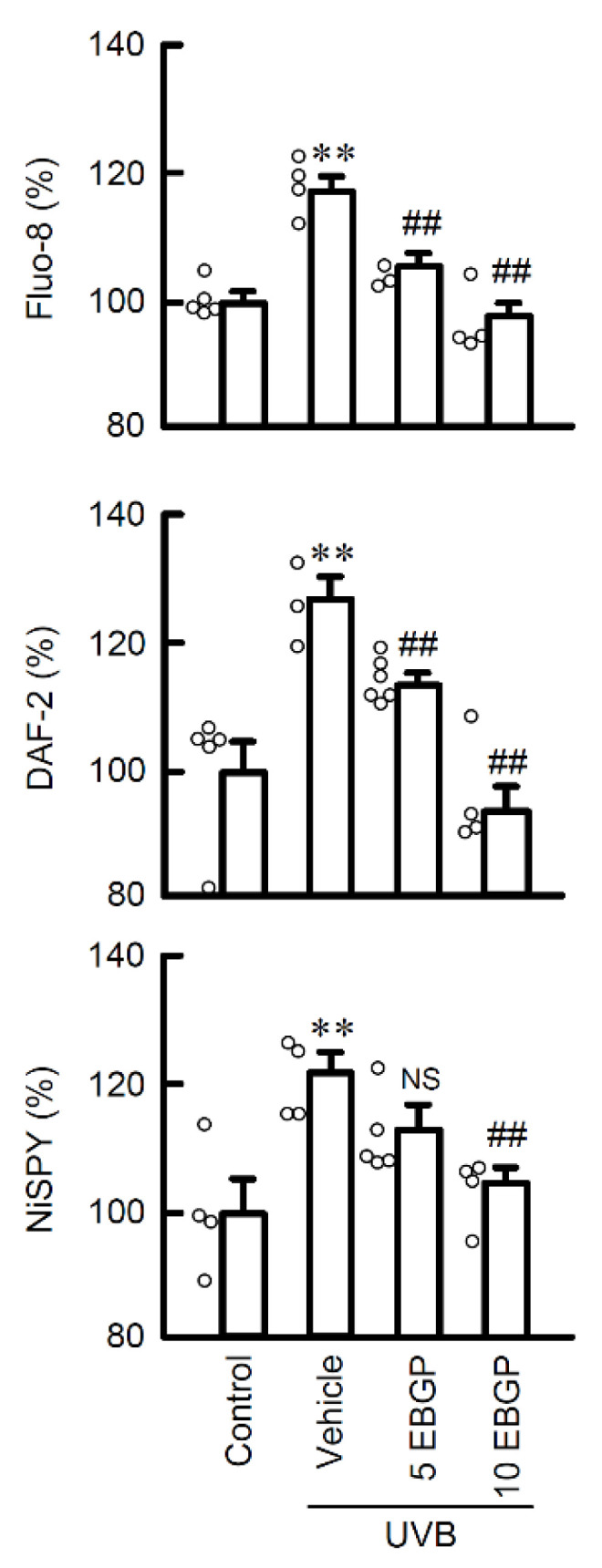
Effect of EBGP on Ca^2+^ influx and RNS generations in the weak UVB-exposed cells. After exposure to weak UVB, the cells were cultured for 3 h. The cells were incubated with 5 μM Fluo-8 AM, 5 μM DAF-2DA, or 5 μM NiSPY-3 for 30 min. Nonirradiated cells were used as control. The fluorescence intensities of Fluo-8, DAF-2, and NiSPY-3 were measured using a microplate reader. *n* = 3–6. ** *p* < 0.01 vs. control. ^##^ *p* < 0.01 and NS *p* > 0.05 vs. vehicle.

**Figure 3 ijms-22-10326-f003:**
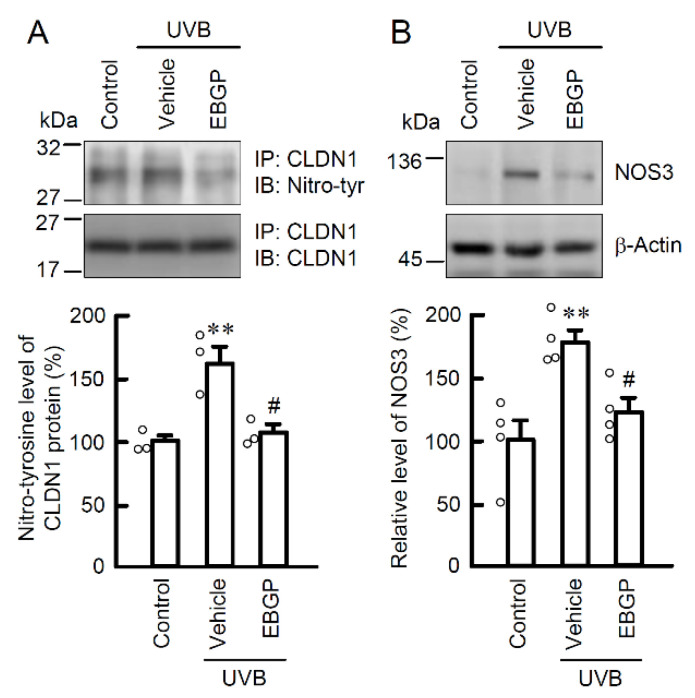
Effect of EBGP on nitratated-CLDN1 and NOS3 expression in the weak UVB-exposed cells. The cells were incubated in the absence (vehicle) and presence of 10 μg/mL EBGP for 30 min, followed by exposure to 5 mJ/cm^2^ UVB for 6 h. (**A**) The cell lysates were immunoprecipitated with anti-CLDN1 antibody. The immunoprecipitants were applied to SDS-PAGE and detected using anti-tyrosine nitration (nitro-tyr) and anti-CLDN1 antibodies. The levels of tyrosine nitration of CLDN1 were represented as a percentage of control. *n* = 3–4. ** *p* < 0.01 vs. control. ^#^ *p* < 0.05 vs. vehicle. (**B**) The cell lysates were applied to SDS-PAGE and detected using anti-NOS3 and anti-β-actin antibodies. β-actin was used as a loading control. The levels of NOS3 expression were represented as a percentage of control. *n* = 4. ** *p* < 0.01 vs. control. ^#^ *p* < 0.05 vs. vehicle.

**Figure 4 ijms-22-10326-f004:**
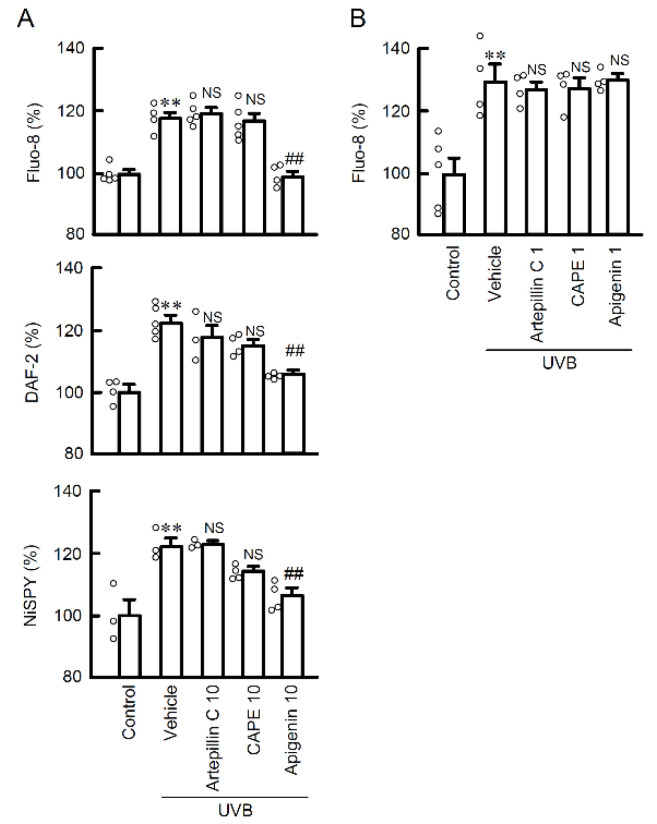
Effects of propolis components on Ca^2+^ influx and RNS generations in the weak UVB exposed cells. (**A**) The cells were incubated in the absence (vehicle) and presence of 10 μM artepillin C, 10 μM CAPE, and 10 μM apigenin for 30 min, followed by exposure to 5 mJ/cm^2^ UVB for 6 h. After 3 h, the cells were incubated with 5 μM Fluo-8 AM, 5 μM DAF-2DA, or 5 μM NiSPY-3 for 30 min. The fluorescence intensities of Fluo-8, DAF-2, and NiSPY-3 were measured by a microplate reader. (**B**) The cells were incubated in the absence (vehicle) and presence of 1 μM artepillin C, 1 μM CAPE, and 1 μM apigenin. The data were presented as a percentage of control. *n* = 3–5. ** *p* < 0.01 vs. control. ^##^ *p* < 0.01 and NS *p* > 0.05 vs. vehicle.

**Figure 5 ijms-22-10326-f005:**
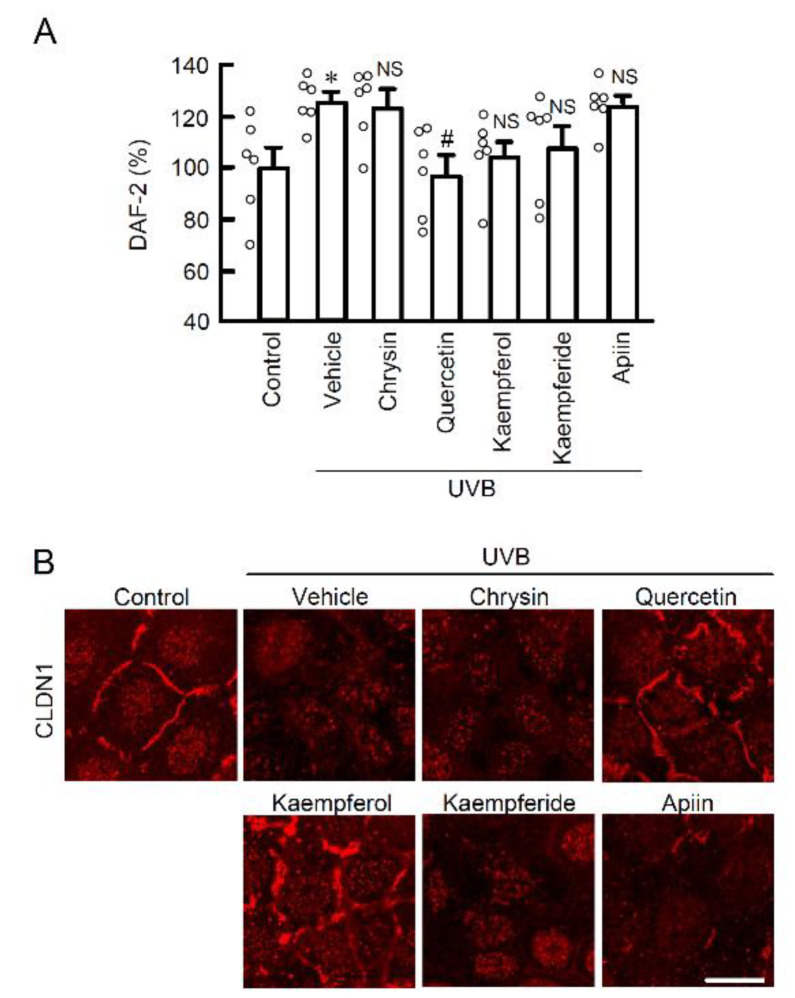
The effects of apigenin-like flavonoids on NO production and localization of CLDN1 in the weak UVB exposed cells. (**A**) After exposing the cells to weak UVB, the cells were incubated for 3 h in the absence (vehicle) and presence of 10 μM chrysin, 10 μM quercetin, 10 μM kaempferol, 10 μM kaempferide, or 10 μM apiin. The cells were incubated with 5 μM DAF-2DA for 30 min. The fluorescence intensity of DAF-2 was measured using a fluorescent microplate reader. *n* = 6. **p* < 0.05 vs. control. ^#^ *p* < 0.01 and NS *p* > 0.05 vs. vehicle. (**B**) Representative images of immunocytochemistry using anti-CLDN1 antibody. Scale bar indicates 10 μm.

**Figure 6 ijms-22-10326-f006:**
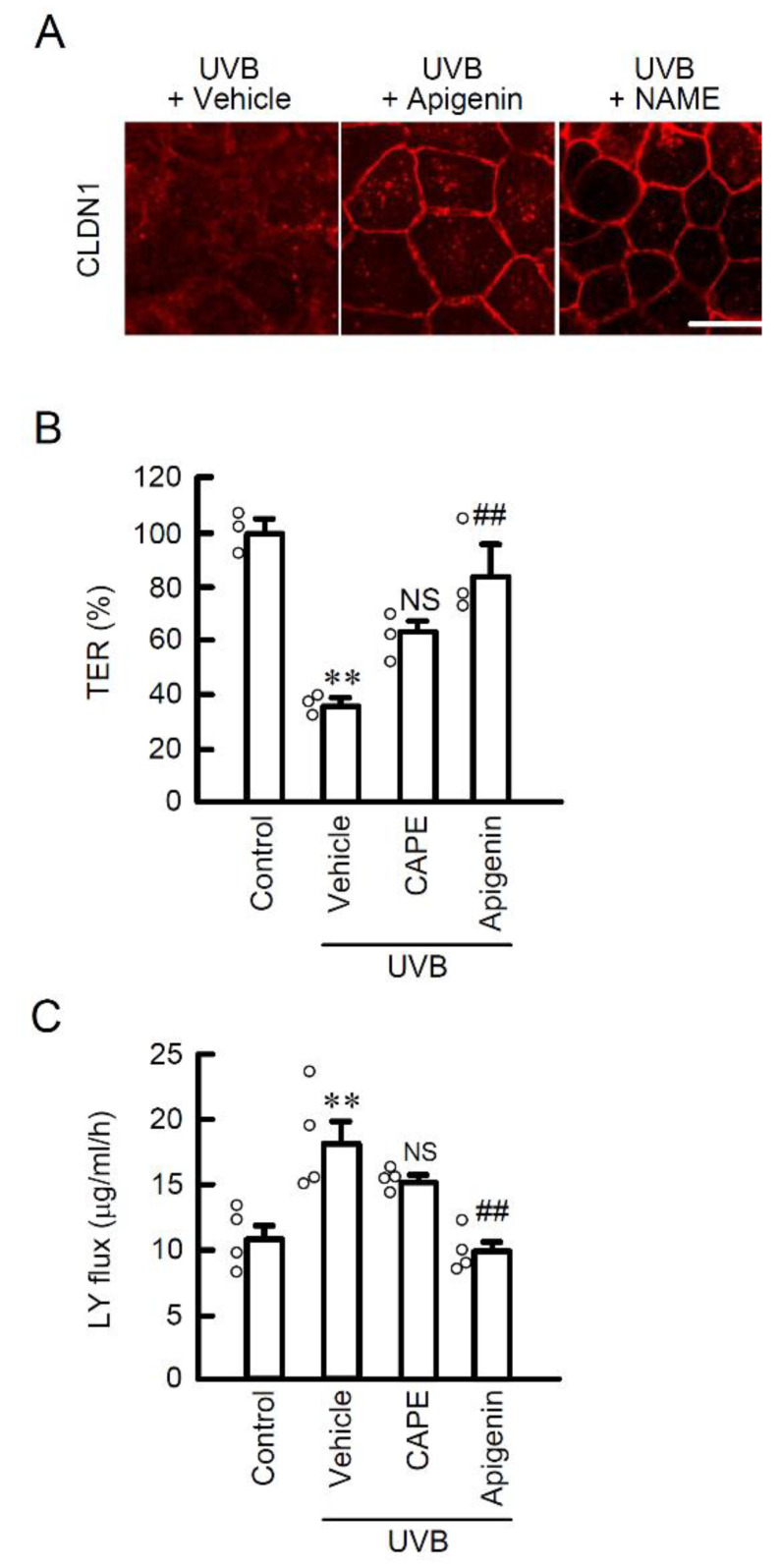
Effect of apigenin on TJ permeability and localization of CLDN1 in the weak UVB exposed cells. (**A**) Representative images of immunocytochemistry using anti-CLDN1 antibody. After exposure to weak UVB, the cells were cultured for 6 h in the absence (vehicle) and presence of 10 μM apigenin, or 100 μM L-NAME. Scale bar indicates 10 μm. (**B**,**C**) After exposure to weak UVB, the cell cultures on transwell plates were incubated for 6 h with 10 μM CAPE, or 10 μM apigenin. (**B**) TER was measured by volt-ohm meter. (**C**) Paracellular permeation of LY was detected using fluorescence spectrometry. *n* = 3–4. ** *p* < 0.01 vs. control. ^##^ *p* < 0.01 and NS *p* > 0.05 vs. vehicle.

**Figure 7 ijms-22-10326-f007:**
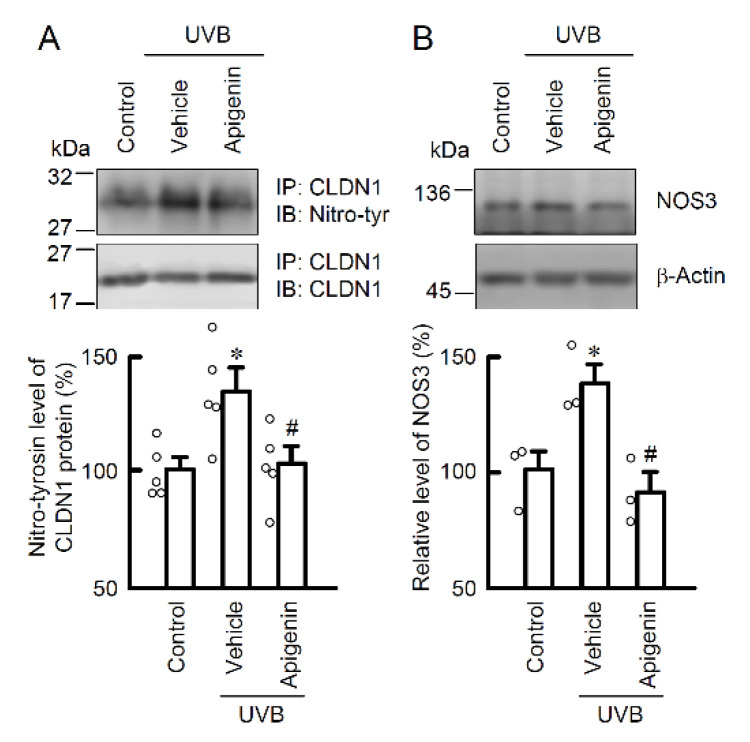
Effect of apigenin on nitration of CLDN1 in the weak UVB-exposed cells. The cells were incubated in the absence (vehicle) and presence of 10 μM apigenin for 30 min, followed by exposure to 5 mJ/cm^2^ UVB for 6 h. (**A**) The cell lysates were immunoprecipitated with anti-CLDN1 antibody. The immunoprecipitants were applied to SDS-PAGE and detected using anti-tyrosine nitration (nitro-tyr) and anti-CLDN1 antibodies. The levels of tyrosine nitration of CLDN1 were represented as a percentage of control. *n* = 3–5. * *p* < 0.05 vs. control. ^#^ *p* < 0.05 vs. vehicle. (**B**) The cell lysates were applied to SDS-PAGE and detected using anti-NOS3 and anti-β-actin antibodies. β-Actin was used as a loading control. The levels of NOS3 were represented as a percentage of control. *n* = 3. * *p* < 0.05 vs. control. ^#^ *p* < 0.05 vs. vehicle.

**Figure 8 ijms-22-10326-f008:**
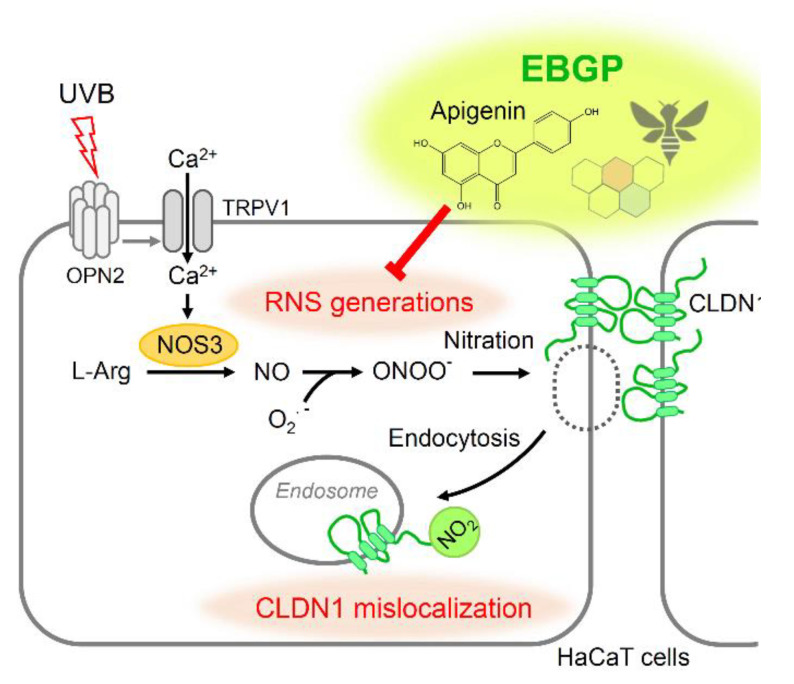
A proposed scheme for the protective effect of EBGP and flavonoids against weak UVB-induced barrier dysfunction in HaCaT cells.

**Figure 9 ijms-22-10326-f009:**
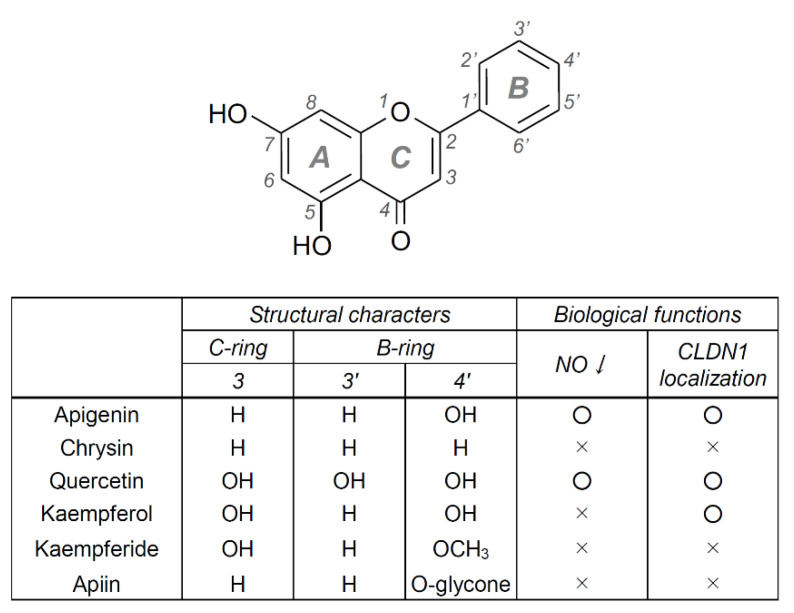
Structural characters and biological function of apigenin-like flavonoids. Representative chemical structure of 5,7-dihydroxylated flavonoid is shown above. The groups at each position on the B- or C-ring in the flavonoid are presented. H, hydrogen; OH, hydroxyl group; OCH_3_, methoxy group; O-glycone, glycosyl substitution of the hydroxy group. The effects of apigenin and apigenin-like flavonoids are summarized in the biological functions column.

## Data Availability

Not applicable.

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
