# Peer review of "Protective Effects of Ethanol Extract of Brazilian Green Propolis and Apigenin against Weak Ultraviolet Ray-B-Induced Barrier Dysfunction via Suppressing Nitric Oxide Production and Mislocalization of Claudin-1 in HaCaT Cells"

_ijms, 2021, doi:10.3390/ijms221910326_

Round 1
Reviewer 1 Report
In this study, the authors revealed that an ethanol extract of Brazilian green propolis (EBGP) show an protective effect against UVB irradiation-induced barrier dysfunction in HaCaT cells. Although the results are reasonable, I have several points that should be addressed by the authors. Specific comments are as follows.
Major points.
- I think the term “weak” UVB is ambiguous. Does this UVB dose (5mJ/cm2) have cytocidal effects or induce growth inhibition on HaCaT cells in 24 to 48 hours? The authors should present the results of cell viability and cell death assay.
- The authors revealed protective effects of some of propolis components (artepillin C, CAPE, and apigenin) against UVB in the results. How much does these components contain in EGBP? This point should be verified.
- Figure 4 to 6: The authors should use the same amount of these components as EGBP (10 μg/mL) and directly compare each other.
Minor points.
- The authors should explain how to dissolve EBGP in materials and methods. What is vehicle in the results?
- Figure 5: I see the UVB treatment dramatically reduced the fluorescence of CLDN1. Is there any change in the amount of the protein?
- The authors should indicate the position of a molecular weight marker above and below the band(s) of interest in western blotting.
- English should be carefully revised by a native English speaker or a professional editing service.
Author Response
Major points
Comment 1
I think the term “weak” UVB is ambiguous. Does this UVB dose (5mJ/cm2) have cytocidal effects or induce growth inhibition on HaCaT cells in 24 to 48 hours? The authors should present the results of cell viability and cell death assay.
Answer
We previously reported that the effect of UVB on the cell viability at 24 h (HaCaT cells, control;100%, 5 mJ/cm2; 100%, 10 mJ/cm2; 90%, 50 mJ/cm2; 60%) (Marunaka et al., IJMS, 2019), indicating that 5 mJ/cm2 is non-cytotoxic dose. Therefore, the following sentence was added to Methods section. Please see lines 306-309, page 10.
Since the cell viability was not affected by 5 mJ/cm2 UVB irradiation at 24 h in previous study [4], this non-cytotoxic dose was used as “weak” UVB.
Comment 2
The authors revealed protective effects of some of propolis components (artepillin C, CAPE, and apigenin) against UVB in the results. How much does these components contain in EGBP? This point should be verified.
Answer
So far, the contents of phenolic and flavonoid compounds in green propolis were reported (artepillin C; 4.80 ± 0.03 mg/g, CAPE; 0.29 ± 0.02 mg/g, apigenin; 0.01 ± 0.00 mg/g) (Andrade et al., 2017). The concentration of each compound used in this study was lower than its content in EBGP. Thus, the potent protective effect of EBGP might be due to the combined effect of these components. We have added following sentences about contents of propolis components in green propolis. Please see lines 263-270, page 9.
Comment 3
Figure 4 to 6: The authors should use the same amount of these components as EGBP (10 μg/mL) and directly compare each other.
Answer
As described above, Andrade et al. have reported the contents of phenolic and flavonoid compounds in green propolis by ultra-high performance liquid chromatographic system coupled with tandem mass spectrometry (artepillin C; 4.80 ± 0.03 mg/g, CAPE; 0.29 ± 0.02 mg/g, apigenin; 0.01 ± 0.00 mg/g) [26]. The concentration of EBGP (10 μg/mL) was equally to ap-proximately 0.16 μM of artepillin C, 0.016 μM of CAPE, and 0.0037 μM of apigenin, respectively. Although the content of flavonoids such as apigenin in EBGP is low, EBGP contains a wide variety of flavonoids [26]. Thus, the potent protective effect of EBGP might be due to the combined effect of flavonoids with characteristic chemical structures.
To compare between EBGP and components, the effects of propolis components at lower dose were examined.
Following sentences and Figure 4B were added to Results section. Please see lines 136-140, page 4.
Lower dose (1 μM) of propolis components did not protect cells against UVB-induced Ca2+ flux. (Fig. 4B). The contents of compounds in EBGP (10 μg/mL) are less than 1 μM of compound [26]. Therefore, the protective effect of EBGP might be derived from combined effects, but not the effect of single compound.
Minor points.
Comment 4
The authors should explain how to dissolve EBGP in materials and methods. What is vehicle in the results?
Answer
According to your, we have added the following sentences in the section of Cell Cultures. Please see lines 298-302, pages 9-10.
EBGP stock solution (10 mg/mL in 100% ethanol) was dissolved in DMEM to prepare the experimental medium (5, 10 μg/mL). As the vehicle, 0.1% ethanol contained DMEM was used. In the other experiments, we used 0.1% dimethyl sulfoxide as the vehicle of propolis components and the other reagents.
Comment 5
Figure 5: I see the UVB treatment dramatically reduced the fluorescence of CLDN1. Is there any change in the amount of the protein?
Answer
We previously demonstrated that 5mJ/cm2 UVB irradiation dose not affect the protein expression levels of CLDN1 in 6 h. (Marunaka, 2019). Therefore, we have added the following sentence in the Results. Please see line172-173, page 6.
In our previous study, weak UVB irradiation induced mislocalization of CLDN1, but not decrease the expression level of protein (Kobayashi, 2020).
Comment 6
The authors should indicate the position of a molecular weight marker above and below the band(s) of interest in western blotting.
Answer
According to your suggestion, we have added the information about molecular weight in the images. Please see new Figures 3 and 7.
Comment 7
English should be carefully revised by a native English speaker or a professional editing service.
Answer
We have re-checked English carefully in the revised manuscript.
Reviewer 2 Report
The manuscript by Yoshino et al. investigated the effects of ethanol extract of Brazilian green propolis and apigenin on HaCAt cells upon UVB irradiation. Although the experimental flows and results sound nicely arranged, this reviewer has the following comments for authors' information to improve their manuscript.
- The introduction seems not sufficient for readers to understand the logic and focus of this study. A more comprehensive review on the natural compounds which have protective effects against UVB should be further added. Another concern is the Brazilian green propolis as the material. The author should explain more to clarify why they used this specific propolis.
- The explanation of the methodology is too simple, which is not helpful at all. Taking UVB irradiation for example, cell culture condition, cell density, and irradiation time are missing. Also, check other method parts.
- It is better to use a scatter dot plot for all figures or make all values visible as dot/circle.
- MW should be added to WB results.
- What is the possible/predicted pathway (mechanism) involved in this protection effect? An interaction or flow should be provided as this last figure.
- Instead of an in vitro cell culture model, the authors should provide evidence if they want to make such a statement like the last sentence of the main text. Alternatively, it is better to discuss if some products show true effects using similar compounds.
Author Response
Response to comments (Reviewer 2)
Major points
Comment 1
The introduction seems not sufficient for readers to understand the logic and focus of this study. A more comprehensive review on the natural compounds which have protective effects against UVB should be further added. Another concern is the Brazilian green propolis as the material. The author should explain more to clarify why they used this specific propolis.
Answer
Since, the protective effect for barrier function was reported by only the green propolis among propolis, we used EBGP in this study. We have added the following sentences about the benefits of propolis and green propolis in the Introduction. Please see lines 53-58 and 62-64, page 2.
In recent decades, a number of research showed that natural product derived from propo-lis have potential benefits on skin, such as protective effect against UV-induced skin damage or sunburn by potent anti-oxidative and anti-inflammatory effects [8, 9]. Therefore, propolis products are attracted much attention as useful candidates for skin health care. Therefore, propolis products are attracted much attention as useful candidates for skin health care.
Moreover, we recently reported that EBGP has a protective effect against oxidative stress-induced skin barrier dysfunction on human keratinocyte derived cells via its potent anti-oxidative effect [4], suggesting the utility ingredient for skin health. (p2, lines 61-63)
Comment 2
The explanation of the methodology is too simple, which is not helpful at all. Taking UVB irradiation for example, cell culture condition, cell density, and irradiation time are missing. Also, check other method parts.
Answer
According to your suggestion, we have added the sentences to explain the detail methodology in the Methods.
Comment 3
It is better to use a scatter dot plot for all figures or make all values visible as dot/circle.
Answer
According to your suggestion, we have added circles plot on all figures to visualize all values. Please see new Figures 1–7.
Comment 4
MW should be added to WB results.
Answer
According to your suggestion, we have added the information about molecular weight in the images. Please see new Figures 3 and 7.
Comment 5
What is the possible/predicted pathway (mechanism) involved in this protection effect? An interaction or flow should be provided as this last figure.
Answer
According to your suggestion, a schematic flow image for the possible mechanism underlying the protective effect has been added as Figure 9.
Comment 6
Instead of an in vitro cell culture model, the authors should provide evidence if they want to make such a statement like the last sentence of the main text. Alternatively, it is better to discuss if some products show true effects using similar compounds.
Answer
Since the present data were in vitro study, we have modified the last sentence in the Abstract and Discussion. Please see lines 26-28, page 1, and lines 279-281, page 9.
Round 2
Reviewer 1 Report
None.
Reviewer 2 Report
This reviewer has no additional comments on this revised manuscript.